# Ventilation during COVID-19 in a school for students with intellectual and developmental disabilities (IDD)

Martin S. Zand [1,2,3]*, Samantha Spallina [4], Alexis Ross [5], Karen Zandi [5‡], Anne Pawlowski [5‡], Christopher L. Seplaki [3,6], Jonathan Herington [7], Anthony M. Corbett [2], Kimberly Kaukeinen [2], Jeanne Holden-Wiltse [2], Edward G. Freedman [4], Lisette Alcantara [2], Dongmei Li [2], Andrew Cameron [8], Nicole Beaumont [4], Ann Dozier [3], Stephen Dewhurst [9], John J. Foxe [4]

1 Department of Medicine, Division of Nephrology, University of Rochester, Rochester, NY, United States of America, 2 Clinical and Translational Science Institute, University of Rochester, Rochester, NY, United States of America, 3 Department of Public Health Sciences, University of Rochester, Rochester, NY, United States of America, 4 The Frederick J. and Marion A. Schindler Cognitive Neurophysiology Laboratory, The Ernest J. Del Monte Institute for Neuroscience, Department of Neuroscience, University of Rochester School of Medicine and Dentistry, Rochester, NY, United States of America, 5 The Mary Cariola Center, Rochester, NY, United States of America, 6 Department of Psychiatry, University of Rochester, Rochester, NY, United States of America, 7 Department of Health Humanities and Bioethics, University of Rochester, Rochester, NY, United States of America, 8 Department of Pathology and Laboratory Medicine, University of Rochester, Rochester, NY, United States of America, 9 Department of Microbiology and Immunology, University of Rochester, Rochester, NY, United States of America

☯ These authors contributed equally to this work.
‡ KZ, and AP also contributed equally to this work.
* martin_zand@urmc.rochester.edu

**Data Availability Statement:** All relevant data are within the manuscript and its Supporting information files.

## Abstract

### Background

This study examined the correlation of classroom ventilation (air exchanges per hour (ACH)) and exposure to $CO_2 \geq 1,000$ ppm with the incidence of SARS-CoV-2 over a 20-month period in a specialized school for students with intellectual and developmental disabilities (IDD). These students were at a higher risk of respiratory infection from SARS-CoV-2 due to challenges in tolerating mitigation measures (e.g. masking). One in-school measure proposed to help mitigate the risk of SARS-CoV-2 infection in schools is increased ventilation.

### Methods

We established a community-engaged research partnership between the University of Rochester and the Mary Cariola Center school for students with IDD. Ambient $CO_2$ levels were measured in 100 school rooms, and air changes per hour (ACH) were calculated. The number of SARS-CoV-2 cases for each room was collected over 20 months.

### Results

97% of rooms had an estimated ACH $\leq 4.0$, with 7% having $CO_2$ levels $\geq 2,000$ ppm for up to 3 hours per school day. A statistically significant correlation was found between the time

**Funding:** Research reported in this Rapid Acceleration of Diagnostics – Underserved Populations (RADx-UP- https://radx-up.org/) publication was supported by the National Institutes of Health under OT2 HD107553 (JF, MZ, SD, SS, AR, KZ, AP, CS, JH, AC, EF, LA, NB, AD, AC}. This work was also supported by the University of Rochester Intellectual and Developmental Disabilities Research Center- P50 HD103536 (JF, MZ, SD, EF, NB, KK, JHW) from the NIH Eunice Kennedy Shriver Institute of Child Health and Human Development- https://www.nichd.nih.gov/, and the University of Rochester Clinical and Translational Science Award UL1 TR002001 (MZ, JHW, KK, AC, AD) from the National Center for Advancing Translational Sciences of the National Institutes of Health-https://ncats.nih.gov/. The content is solely the responsibility of the authors and does not necessarily represent the official views of the National Institutes of Health. The funders had no role in study design, data collection, and analysis, decision to publish, or preparation of the manuscript.

**Competing interests:** The authors have declared that no competing interests exist.

that a room had $CO_2$ levels $\geq 1,000$ ppm and SARS-CoV-2 PCR tests normalized to room occupancy, accounting for 43% of the variance. No statistically significant correlation was found for room ACH and per-room SARS-CoV-2 cases. Rooms with ventilation systems using MERV-13 filters had lower SARS-CoV-2-positive PCR counts. These findings led to ongoing efforts to upgrade the ventilation systems in this community-engaged research project.

## Conclusions

There was a statistically significant correlation between the total time of room $CO_2$ concentrations $\geq 1,000$ and SARS-CoV-2 cases in an IDD school. Merv-13 filters appear to decrease the incidence of SARS-CoV-2 infection. This research partnership identified areas for improving in-school ventilation.

## Introduction

A variety of mitigation measures have been employed to prevent SARS-CoV-2 exposure and infection, including immunologic (e.g. vaccination, monoclonal antibodies), antiviral (e.g. paxlovid), and isolation methods (e.g. masking, physical distancing, stay-at-home policies). A key approach to environmental mitigation has been the use of high volume ventilation of enclosed spaces (i.e. adequate flow of fresh, uncontaminated air), which has been shown to decrease the risk of infection from a variety of respiratory pathogens [1–3], and this effect has been widely presumed to include SARS-CoV-2. Schools were an immediate focus in many areas with respect to mitigation measures for SARS-CoV-2, including potentially costly recommendations for improving ventilation. However, very little data are available assessing the actual impact of ventilation on in-school transmission of SARS-CoV-2.

The SARS-CoV-2 $\beta$-coronavirus is spread primarily via airborne droplets and aerosols, which can remain circulating for up to three hours [4]. Ventilation is defined as the process of supplying clean air to the indoor air of a dwelling by natural (e.g., open windows if the outdoor air is sufficiently clean) or mechanical means (e.g. heating, ventilation, and air conditioning; HVAC) often involving recirculating the air after cleaning with a filter [5]. Increasing ventilation is one means of reducing exposure to airborne viral pathogens, and increasing ventilation rates and upgrading inline HVAC particle filters (e.g. from MERV-8 to MERV-13) were widely recommended for public venues including businesses, transportation, and schools [6–8]. For schools specifically, improving ventilation as a risk reduction strategy was the subject of many recommendations [8–11]. However, very little robust data are available assessing the actual impact of improved ventilation on in-school transmission of SARS-CoV-2. One study found 74% lower aggregate rates of COVID-19 in schools with mechanical versus natural ventilation [12]. While there have been reports of outbreaks within schools [13, 14], a number of epidemiological studies have concluded that students and school staff had relatively low risk of SARS-CoV-2 in-school transmission [15–17] and schools were unlikely to be a major factor in the pandemic spread [18].

Despite the larger number of epidemiological studies in public schools, we are not aware of similar studies in schools for students with intellectual and developmental disabilities (IDD), who face greater risks [19–24], and impacts from, infection. The majority of students with IDD are served by IDD-specialized schools that provide educational services, comprehensive

clinical services, speech and psychological interventions, nutritional needs, and physical care. Staff-to-student ratios are much higher than in typical K-12 schools, with more frequent, sustained and close physical contact between staff and students [25]. Students with an IDD are at higher risk of exposure to SARS-CoV-2 [25], severe infection, and mortality [24]. The IDD spectrum covers many conditions (e.g. autism spectrum disorder, genetic disorders), which are associated with immune abnormalities that increase risk of SARS-CoV-2 and poor vaccine responses [19–24]. Many students with an IDD have difficulty with conventional infection prevention measures such as masking and social distancing. This also puts staff in IDD settings at higher risk of exposure to SARS-CoV-2 from asymptomatic but infected students. Conversely, the impact on IDD students, and their caregivers, of school closures, isolation and quarantine is heightened [26]. IDD students receive many specialized services (e.g. physical and speech therapy) in school that are not readily available at home. Thus, it is essential to identify interventions that are implementable in the unique environment of IDD-specialized schools that retain students in school while mitigating the risk of SARS-CoV-2 infection. Increased ventilation has been proposed as one such measure.

Time series measurements of $CO_2$ levels have been used as a frequent surrogate for ventilation adequacy in schools [8, 27–32]. As $CO_2$ is exhaled during respiration, it accumulates in indoor spaces depending on the air exchange rates. The dynamic balance between $CO_2$ production and ventilation with air having a lower $CO_2$ concentration determines the time-varying $CO_2$ concentration within a room. Accordingly, time series measurements of room $CO_2$ concentrations can be used to derive a surrogate marker for ventilation adequacy, typically measured as air changes per hour (ACH) [27]. International guidelines recommend a minimum of 4 ACH for school classrooms, and 8–15 ACH for auditoriums, athletic facilities, cafeterias and other high-density or high-activity enclosed spaces [5]. Mismatched ventilation can result in elevated indoor $CO_2$ levels that can affect learning and cognitive performance [33], areas of key focus in schools for students with an IDD. Unfortunately, many school buildings have outdated HVAC systems, resulting in low or very low ACH [6, 34, 35].

The National Institutes of Health (NIH) program "Rapid Acceleration of Diagnostics for Underserved Populations (RADx-UP)" was designed to study ways to support vulnerable populations during the COVID-19 pandemic. A major goal of the program was to study strategies to keep vulnerable populations of students in school while mitigating the risk of exposure and in-school spread of SARS-CoV-2 among students and staff. Primarily designed to study the utility of screening for infection with frequent molecular testing, our RADx-UP project work has also focused on the effect of a variety of measures on mitigating exposure and infection in an IDD-specialized school setting. Here we report the results of an extensive study of ventilation across 3 buildings and 100 rooms in an IDD school for children and young adults (ages 3–21 years) during the SARS-CoV-2 pandemic.

## Methods

### Human subjects protection

This study was approved by the University of Rochester Medical Center Research Subjects Review Board (RSRB STUDY00005294). Participants provided written informed consent, and parents or guardians provided consent for minors. Recruitment occurred April 1, 2021 and through May 31 2023. All subjects' information and research data were coded in compliance with the Department of Health and Human Services Regulations for the Protection of Human Subjects (45 CFR 46.101(b) (4)).

## Room and building data

The school in this study is composed of 3 different buildings, each with a varying number of classrooms, therapy and activity rooms, gyms, and a variety of offices. We purposefully selected 100 school rooms, from a total of 267, reflecting the different room types in use across the 3 different buildings. Metadata collected included a unique room identifier, building, square footage, ceiling height, total occupancy range, and room type (e.g. classroom, therapy, etc.). In addition, we obtained further data regarding the heating, ventilation, and air conditioning (HVAC) systems used in each building relevant to the study, including type of ventilation (1 versus 2 stage), percentage of outside air mixing, and filtering (S1 File).

## SARS-CoV-2 case data

Total counts of positive SARS-CoV-2 PCR tests were tabulated for the period from August 2021—August 2022 for the rooms where $CO_2$ concentrations were measured. During this period at the school, most individuals spent the majority of the day in a single classroom, and thus positive tests were associated with the primary classroom assignment for any individual. Individuals who tested positive within 30 days of a prior positive PCR were considered single infections.

## CO$_2$ measurements

In November 2022, we measured ambient $CO_2$ levels, temperature, and humidity in 100 school rooms using ARANET4 sensors (Aranet, Aurora Colorado) with firmware version 1.2, with one sensor per classroom, twenty classrooms at a time. Monitors were placed on the ceiling near the air return vents in the center of the room to ensure mixing and avoid dead spaces within each room. In a small number of rooms, where ceiling placement was not possible, monitors were placed on side walls near air return vents. The indicator screens of the devices were covered so that classroom occupants would not alter ventilation (e.g. open windows or doors) outside of their usual pattern. Sensors were set to record ambient $CO_2$, temperature, and humidity levels every 2 minutes continuously for 24–72 hours. Data were downloaded to an iPad (Apple Inc., Cupertino CA) using the Aranet manufacturer's software as a .CSV file. Each record in a .CSV file identified the sensor and time of each measurement. A separate .CSV file identified the school room and sampling start and stop times. Individual .CSV files were then processed with a custom R program that added the unique room ID number to each record based on the sampling start and stop times and the unique sensor ID. Duplicate records were removed. The final data set contained 150,286 individual measurements (S2 File). To assess intra-day variation, some rooms had data continuously recorded for up to 120 hours, both during the normal weekly school day, and over the weekend to assess temperature variations.

## Air changes per hour estimation

To estimate air changes per hour (ACH), $CO_2$ time series for each room (Fig 1) were first subjected to Gaussian blurring to scale $\sigma = 10$ and then local maxima identified using Mathematica (vers 13.1; *FindPeaks*; Wolfram Research). Local minima were similarly identified by analyzing the negative time series, resulting in pairs of $(c_i^{max}, c_i^{min})$, which were then filtered to exclude pairs with small declines in $CO_2$ levels where $c_i^{max} - c_i^{min} < 120$ (2 rooms) [27, 36]. This resulted in 98 rooms with measurements adequate for estimating ACH. We then fitted an

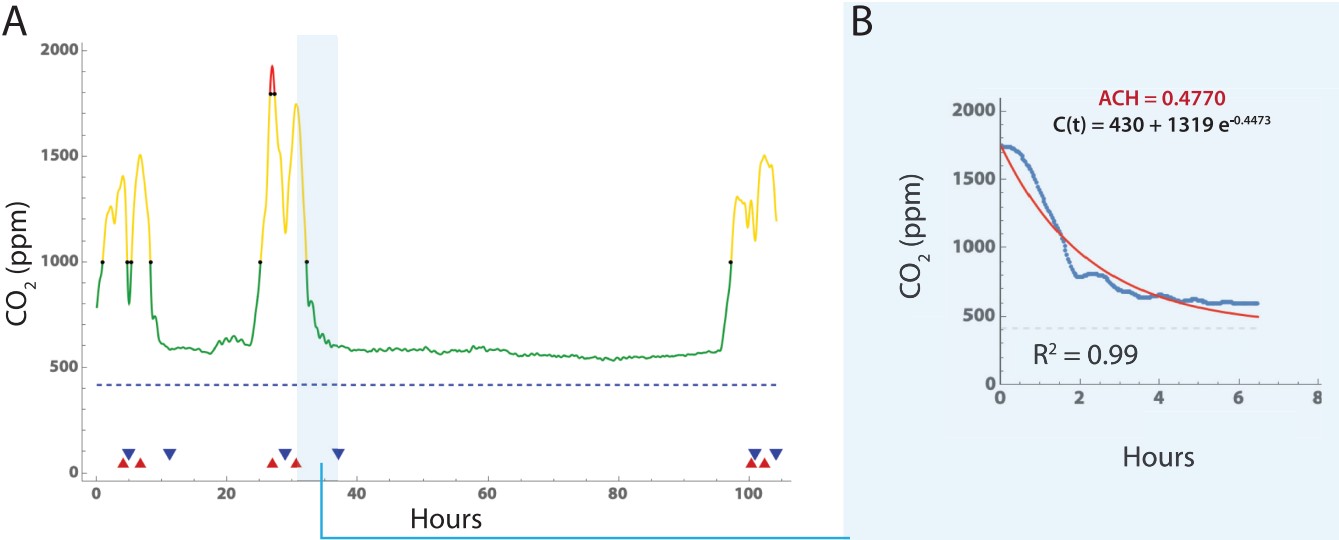

**Fig 1. Estimation of air changes per hour using ambient CO₂ levels.** Example of $CO_2$ time series measurement, peak identification and fitting. A: $CO_2$ levels versus time in curve with Gaussian smoothing window of 10, and peak (red triangle) and valley (blue triangle) identification. Segments with a long tail were truncated at the point where subsequent readings were only 2% lower than the mean of the previous 10 readings. B. Eq 1 was fitted to each set of interval data using *NonlinearModelFit* function in Mathematica, and the Levenberg-Marquardt algorithm option. In those rooms with multiple (peak, valley) segments, the final room ACH value was estimated by averaging the estimates from all segments.

exponential decay curve [27] of the form:

$$c_t = (c_i^{max} - c_R)e^{-\alpha t} + c_R \qquad (1)$$

to each time series segment between successive $(c_i^{max}, c_i^{min})$ pairs, where $i$ = number of segments identified for an individual room, $\alpha$ is air exchanges per hour, and $t$ is the time from $c_i^{max}$ to $c_i^{min}$. Fitting was performed using the *NonlinearModelFit* function in Mathematica with the Levenberg-Marquardt algorithm option (Fig 1). The room airflow needed to achieve 4 air changes per hour was estimated using the volume of the room (cubic feet) divided by 4.

## Statistical analysis

The incidence of positive PCR tests in each room over 12 months was normalized to mean room occupancy from 1 month worth of attendance and occupancy data during a fall semester of the school year in 2022 during the COVID-19 pandemic. Correlation between occupancy normalized counts of positive SARS-CoV-2 tests and air changes per hour (ACH) or exposure time to $CO_2 \geq 1{,}000$ ppm were calculated using the non-parametric Spearman's rank correlation test, with the explained variance reported as the Spearman's $\rho$ statistic.

## Results

### Room characteristics

We sampled 100 rooms spread across three buildings. Two buildings had single stage HVAC systems, which supplied ventilation only when triggered by the need for heating during the sampling period (November), while the third building had a two stage HVAC system. Set-points for heating were set at 70˚F from 6 AM—6 PM, and 65˚F outside those hours. Outside temperatures during the measurement periods were all ≤65˚F. For all HVAC systems, air provided to rooms was 20% outside and 80% recirculated air. One building had HVAC systems

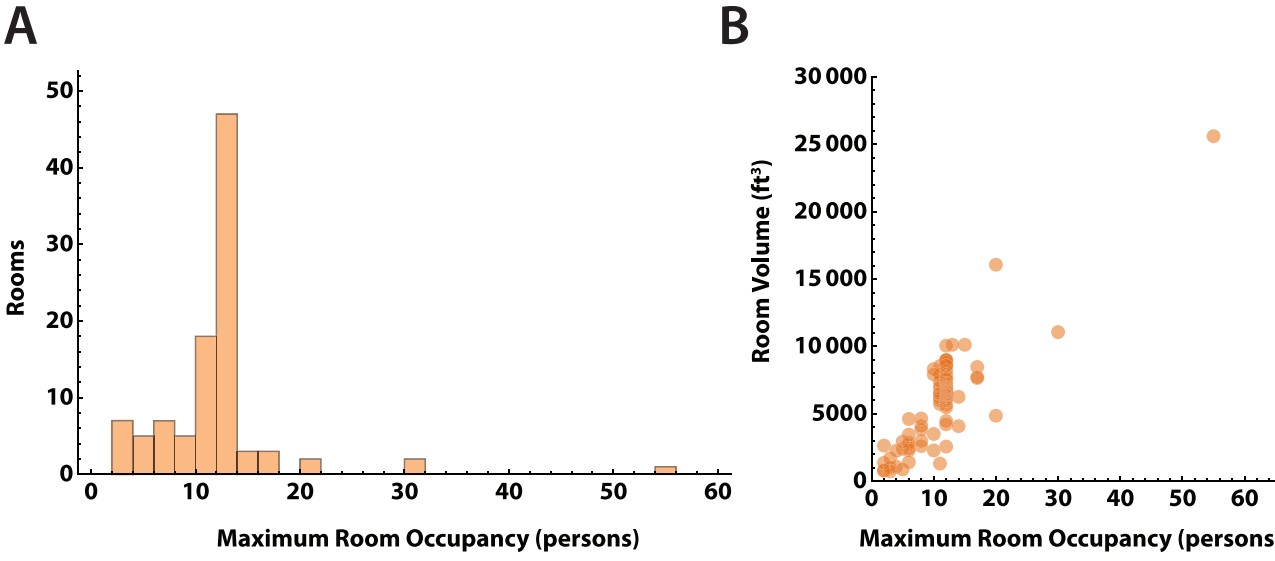

**Fig 2. Capacity and volume of study rooms.** A. Distribution of maximum room occupancy in persons for the 100 rooms in the study. B. Room occupancy versus volume in ft$^3$. Most rooms had ceiling heights of 9–10 feet, and the relationship between room volume and area is close to linear. Larger rooms had higher occupancy levels.

with inline MERV-13 filters installed to mitigate SARS-CoV-2 transmission from recirculating air. The HVAC systems in two other buildings had MERV-11 filters as the blowers were not able to overcome the higher filter resistance from MERV-13 filters. The distribution of maximum room occupancy is shown in Fig 2A. Room dimensions were measured, with mean room area $752 \pm 492$ ft$^2$, and mean ceiling heights $8.6 \pm 2.0$ ft. As expected, peak room occupancy varied linearly with room volume (Fig 2B). Windows remained closed in all classrooms during the measurements.

## CO$_2$ level profiles

Time series measurements of room CO$_2$ for the 98 rooms analyzed showed an expected increase in CO$_2$ concentrations during the day, with a decline overnight to baseline. Several patterns were evident (Fig 3A), including a simple peak-trough, dual peak-troughs, and multiple peak-trough pairs. We then examined if peak CO$_2$ levels in each room could be correlated with room volume (Fig 3B), air changes per hour calculated from each peak-trough pair (Fig 3C), or maximum room occupancy (Fig 3D). No obvious correlations were present, likely due to the variation in ventilation systems and room types among the 3 buildings.

## Room CO$_2$ concentrations

CO$_2$ levels $\geq 2500$ ppm have been shown to decrease cognitive task performance in neurotypical adults on standard test instruments [37, 38]. We were therefore interested in the amount of time each room's CO$_2$ concentrations were within four different ranges of CO$_2$ during an 8 hour school day (7AM—3PM). Several patterns were evident (Fig 4. Rooms in Group 1 (n = 18) remained within the recommended values of CO$_2$ ($\leq 1000$ ppm) [5] throughout the entire period. In contrast rooms in Group 2 (n = 74) had CO$_2$ levels of 1001–2000 ppm, which have been shown to cause drowsiness and modest decreases in performance of cognitive tasks in neurotypical adults [37, 38] and children [39–42]. Of note, occupants of rooms in Group 3

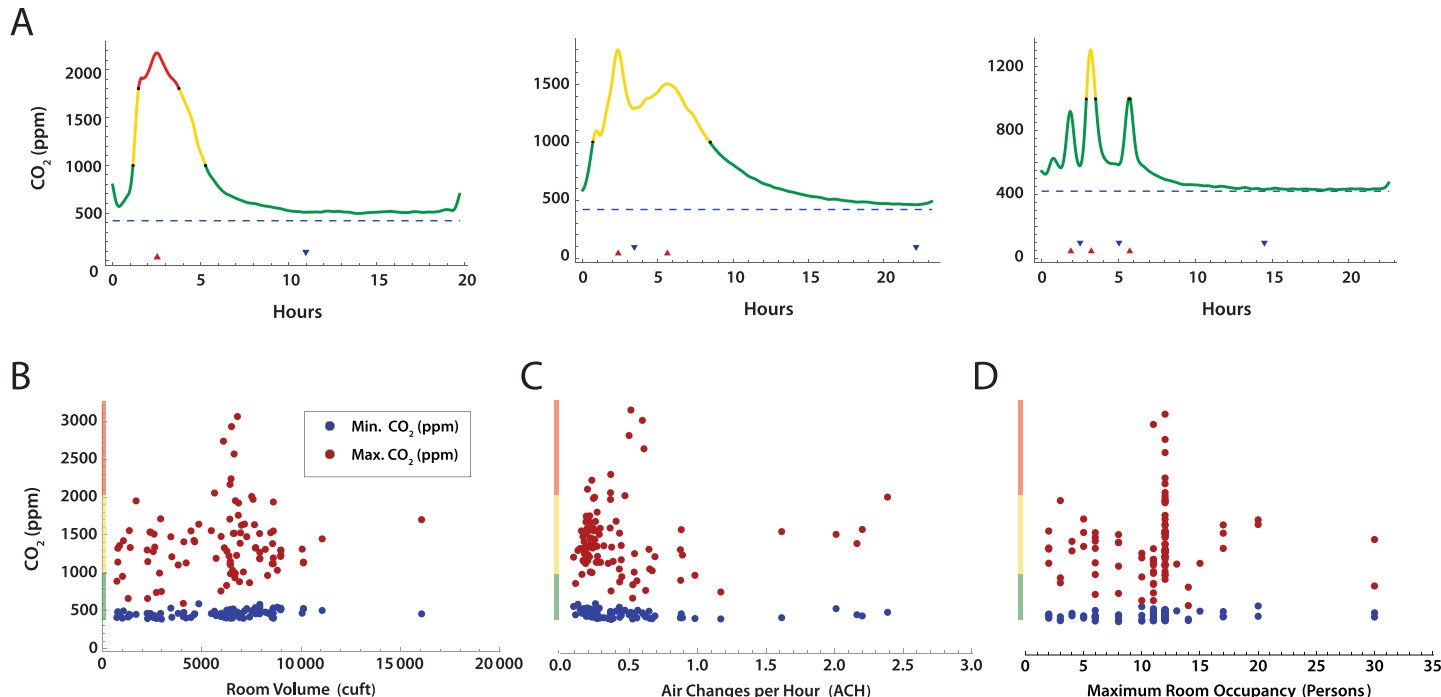

**Fig 3. $CO_2$ concentration measurements over time in school rooms.** A. Selected patterns of $CO_2$ concentration over the school day. Color coding indicates $CO_2$ concentration stratification by concentration bands commonly used by the American Society of Heating, Refrigerating and Cooling Engineers (ASHRAE) (green, < 1,000 ppm; yellow 1,000—2,000 ppm; red, ≥ 2,000 ppm). Minimum (blue) and maximum (red) $CO_2$ concentration versus B. Room volume, C. Estimated ACH, and D. Maximum room occupancy.

(n = 6; 2001–3000 ppm) and Group 4 (n = 2; ≥3000 ppm) had approximately 1–6 hours at moderately high $CO_2$ levels, although still below the OSHA occupational limits of ≤5000 ppm. We confirmed the elevated $CO_2$ levels in Group 3 and 4 rooms with 3 more full day time series measurements.

We also investigated whether the maximum $CO_2$ concentrations observed in each room correlated with functional room type and building. Several different functional room types were present among the 3 buildings: classroom, therapy, treatment, activity, gym, office, kitchen, break, copy, music, and technology. Classrooms generally are occupied during the day by 7–8 students, 1 teacher, and 2–4 aids, with activity levels that produce moderate amounts of $CO_2$. In contrast, music, activity, and gym rooms all have activities that may generate larger emissions of $CO_2$. Each building also has a different type and vintage of HVAC system, with potential variation in ventilation. To assess the distribution of peak $CO_2$ values, we sorted rooms by type and building, as shown in Fig 5. Note that building B had the widest distribution of maximum measured $CO_2$ values, and these were found in classrooms. While a number of the measured $CO_2$ concentrations were above the ASHRAE guidelines for classrooms for varying periods of time [5], none exceeded OSHA's exposure limits [43].

## Temperature versus $CO_2$ relationship

The heating HVAC systems for each of the 3 buildings had zone heating and cooling. Two buildings were described as single stage, with mechanical room ventilation occurring only when the room needed to be heated or cooled, while one was described as dual stage, with both a baseline level of ventilation and then increased airflow when heating or cooling was

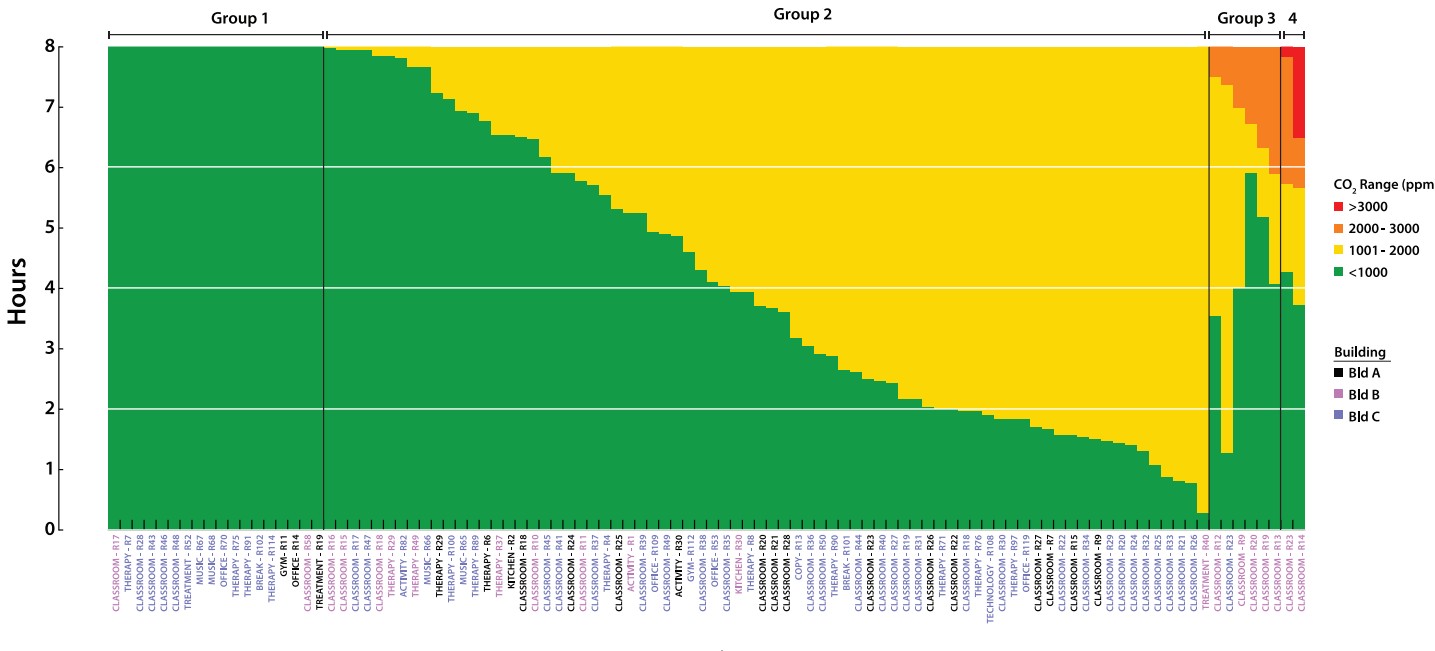

**Fig 4. CO₂ level exposure.** The distribution of $CO_2$ exposure during an 8 hour school day. Stacked plots indicate the amount of time spent at each $CO_2$ level during the school day. Horizontal grid lines indicate quartiles. Each bar represents a room, with labels coded by building. Rooms are labeled by function: classroom, therapy, treatment, activity, gym, office, kitchen, break, copy, music, and technology.

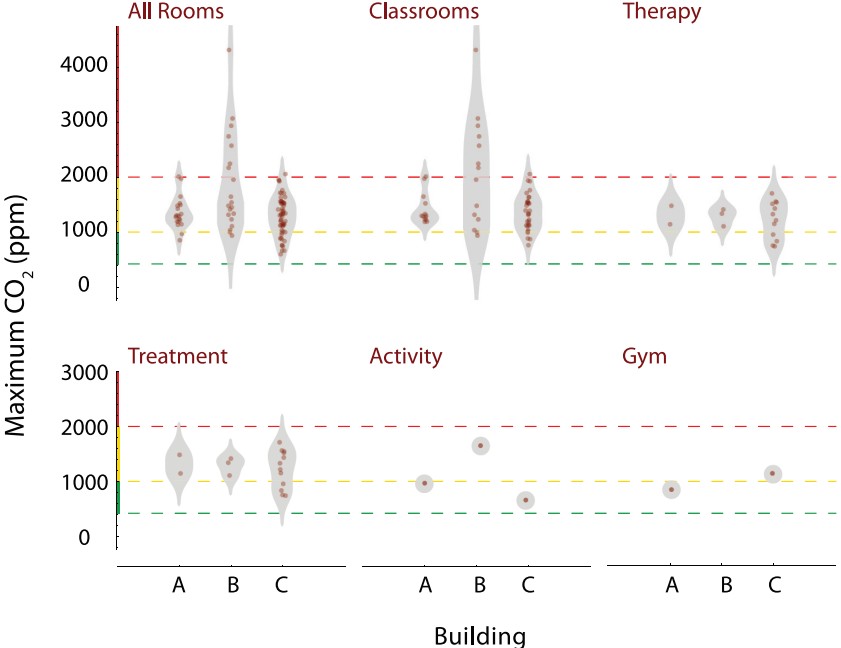

**Fig 5. Peak CO₂ levels by room functional type and building.** Each plot displays a density distribution, along with individual measurements, for the room types indicated. Horizontal grid lines indicate ranges of $CO_2$ which are acceptable (green), moderately elevated (yellow), and high (red). Distributions for room types are further categorized by building.

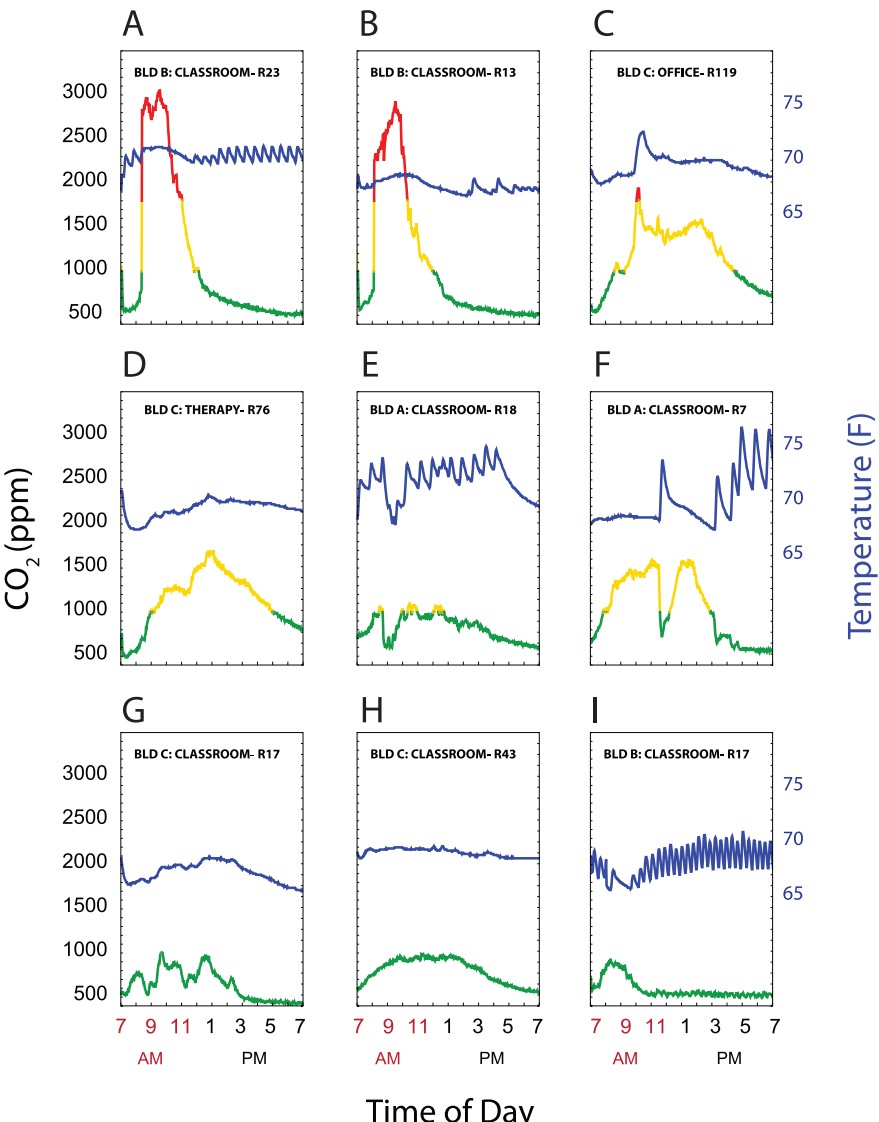

**Fig 6. CO₂ levels and room temperature.** Plots of CO$_2$ levels (left axis) and temperature (right axis) showing representative room patterns. (A-C) show rooms that had CO$_2$ levels $\geq$2000 ppm, (D-F) had CO$_2$ peak levels 1000–2000 ppm, and (G-I) had CO$_2$ $\leq$1000 from 7AM-7PM. CO$_2$ are coded by $\leq$ 1,000 ppm (green), 1,000–2,000 ppm (yellow), and > 2,000 ppm (red). Temperature is represented by the blue line.

required. CO$_2$ values are a function of both ongoing production and ventilation. We hypothesized that room CO$_2$ measurements would decrease during periods when the HVAC system was activated during room occupancy. Fig 6 shows superimposed plots of CO$_2$ levels and temperature readings over time in a representative sample of 9 out of 100 rooms. Note the oscillations in temperature as the heating system engages (e.g. Fig 6A, 6E, 6F and 6I) in some rooms. Of note, Fig 6 panels C,E,F and I all show instances where the temperatures increased rapidly, indicating active mechanical heating and ventilation, with a corresponding decline in room CO$_2$ levels. These findings suggest that, at ambient temperatures above (in winter) or below (in summer) the thermostat set point, HVAC cycling coupled to temperature regulation may

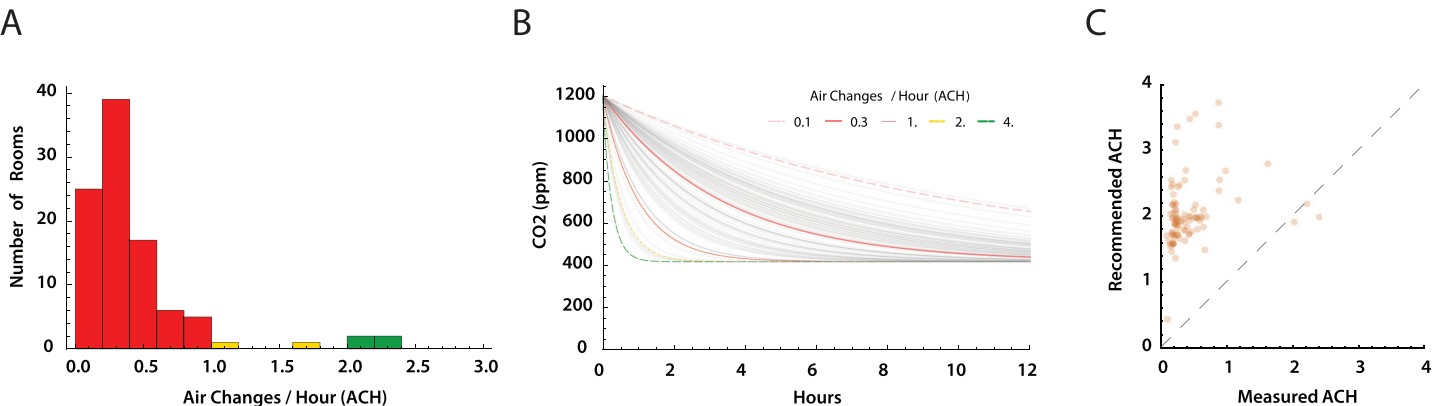

**Fig 7. Estimates of air changes per hour (ACH).** (A) Distribution of estimated ACH. Only four rooms had an estimated ACH $\geq 2.0$. The recommended ACH for schools is $\geq 4.0$ exchanges of fresh air per hour [5, 9]. (B) Each ACH measure was used to model the decrease in $CO_2$ levels over time from a starting concentration of 1200 ppm $CO_2$ (gray lines). Colored reference lines show expected $CO_2$ clearance with 0.1, 0.5, 1.0, 2.0, and 4.0 ACH. (C) Estimated room ACH plotted against recommended ACH for each room based maximum room occupancy as recommended for school rooms American Society of Heating, Refrigerating and Air-Conditioning Engineers (ASHRAE) standard 61.1 calculated from room area and occupancy [5]. Dashed line is where $ACH_{measured} = ACH_{recommended}$.

have decreased $CO_2$ clearance, since the blowers circulate air at higher flow rates only when temperatures fell outside of the thermostat set point.

## Air changes per hour estimates

Using the $CO_2$ level profiles for each room, we derived air changes per hour (ACH) by fitting an exponential decay model to the time series segments where $\Delta CO_2 \geq 120 ppm$ between peak and trough in a decay segment (S1 Fig). The distribution of ACH values is shown in Fig 7A, stratified by very low (red; ACH$\leq 1.0$), low (yellow; ACH 1.0–2.0), and moderately acceptable ACH$\geq 2.0$). All were less than the recommended ACH level of $\geq 4.0$ [5]. Fig 7B demonstrates clearance of a 1200 ppm $CO_2$ load at the estimated ACH for each room. Clearance times of 4–9 hours were consistent with measured, overnight, $CO_2$ level declines. Finally, we compared the measured ACH values for each room with the minimum ACH recommended based on maximal room occupancy and measured room area and volume (Fig 7C. The estimate of recommended ACH based on table 6–1 in the ASHRAE guidelines (Classrooms; ages 5 to 8) [5] require an air flow 10 ft$^3$ min$^{-1}$ per person and 0.12 ft$^3$ min$^{-1}$ per ft$^2$ of area. The air flow was divided by the room volume to obtain recommended ACH. This calculation may underestimate required air flow, and thus recommended ACH, as it does not account for re-mixing of outside (i.e. fresh) air (20%) and recirculated air (80%) used by the HVAC systems.

## Incidence of SARS-CoV-2-positive PCR tests

We next sought to determine if rooms with lower ACH had higher numbers of SARS-CoV-2 infections (Fig 8), and whether MERV-13 filters altered this result. During the COVID-19 pandemic, the use of high efficiency inline particle filters (e.g. MERV-13) were recommended to improve filtration of SARS-CoV-2 within HVAC systems [2, 8–10]. Higher filter efficiency, however, often results in lower ventilation rates and greater strain on the HVAC mechanical blowers such that older HVAC systems may not accommodate MERV-13 filters. In this school, two building HVAC systems could not provide the higher mechanical load required to circulate air through MERV-13 filters, and continued to use MERV-11 filters. We first gathered data for a 1-year period for positive SARS-CoV-2 PCR test counts per classroom for those

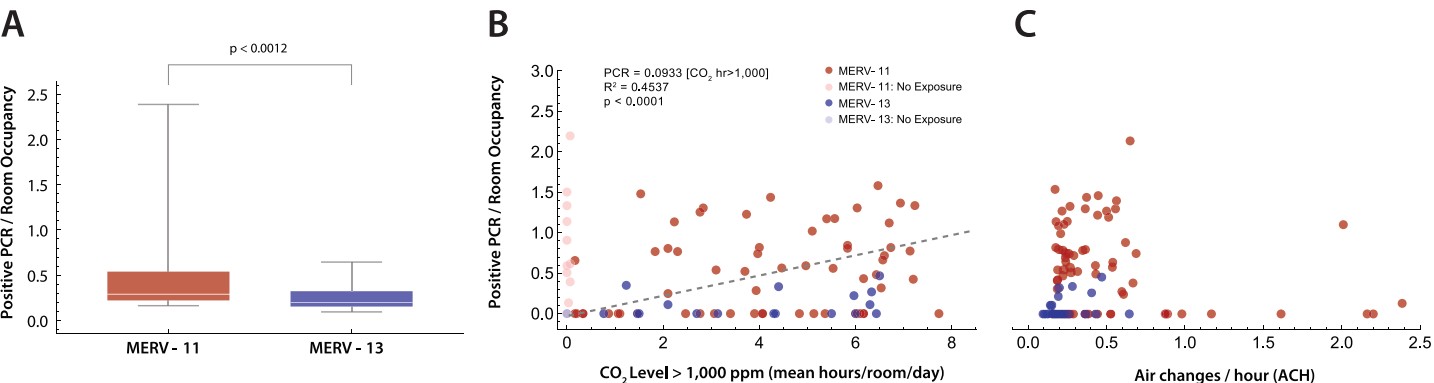

**Fig 8. Positive PCR tests per room.** (A) Difference in PCR tests per room, normalized by room occupancy, between rooms with HVAC systems using MERV-11 (red) versus MERV-13 filters. Statistical comparison with the Mann-Whitney U-test (p<0.0012). (B) The time in each room where the ambient $CO_2 \geq 1,000$ ppm was plotted against positive PCR tests in that room normalized by mean room occupancy. Rooms with $CO_2 \leq 1,000$ ppm for the entire day were excluded (lighter markers). MERV filter status for the building rooms is shown (red, MERV-11; blue, MERV-13). Linear regression (dashed line) had an $R^2 = 0.46$, with an $R^2 = 0.4537$ explaining 45% of the variance. (C) ACH versus positive SARS-CoV-2 PCR tests per room normalized to room occupancy. No statistically significant correlation was found ($R^2 = 0.0036$).

rooms that had measurements of $CO_2$ levels, estimates of ACH, and MERV filter data. Positive PCR tests were normalized to mean classroom occupancy. A comparison of rooms with MERV-11 filters demonstrates that they have a higher occupancy normalized positive PCR tests per room compared to rooms with MERV-13 filters (Fig 8A). Next, total hours of $CO_2$ levels $\geq 1,000$ ppm were tabulated for each room. A significant correlation was found between hours of classroom $CO_2$ levels $\geq 1,000$ ppm and the incidence of SARS-CoV-2 infection (Spearman rank correlation; p = 0.01937, $\rho$ = 0.2864). Lastly, no statistically significant correlation was found between estimated ACH and the incidence of positive PCR tests normalized to mean classroom occupancy.

## Discussion

Our overall findings revealed only a small number of PCR-detected SARS-CoV-2 cases in this IDD school over the course of one year. Our finding that less than 20% of staff and students had PCR-confirmed SARS-CoV-2 infections is in agreement with other school studies. Investigators in Norway [17], Germany [44], Italy [13], United States [45–47] and Ireland [16] found minimal or no evidence for in-school SARS-CoV-2 transmission. Although there have been reports of small secondary infection clusters [14, 48], larger studies have concluded that, overall, schools were unlikely to be a major factor in the pandemic spread [18]. Although one retrospective cohort study in Italy found that schools with mechanical ventilation had a 78% lower incidence of SARS-CoV-2 cases, their findings may have limited applicability in geographic regions where seasonal heating and cooling is necessary. Additionally, the conditions in IDD schools are likely somewhat different than public schools reported in the above publications. In the school studied in this work, only 30% of the students were able to tolerate masking, while all of the professional staff was masked. Thus, further study of the combined effects of mitigation measures is needed.

Using $CO_2$ levels to estimate ACH, we found that the school ventilation systems did not achieve ACH $\geq 4$, which is consistent with other reports [49–52]. However, the use of $CO_2$ measurements to derive ACH comes with several important caveats [53]. To fit exponential decay models to $CO_2$ concentrations, a sufficient decay time is needed after $CO_2$ loading [27, 54]. In this study, the $CO_2$ concentration decay phases occurred at the end of the school day

and overnight. However, the HVAC systems had slightly different temperature set-points at night, which may have led to different ventilation rates, underestimating daytime ACH [27]. In addition, the HVAC systems were set primarily to maintain temperature, and thus higher $CO_2$ concentrations may have been due to lower air circulation when the heat generated by room occupancy decreased the need for higher rates of heated air circulation [27]. Ventilation rates can also vary in both naturally- and mechanically-ventilated buildings, depending the indoor-outdoor temperature difference, wind speed, heating and cooling load, building envelope leakage, and other factors that may change over time [27, 54]. Our measurements were performed in the winter season (November) over 3 contiguous days, when heating was necessary due to outside temperatures, which is relevant to both HVAC use and the respiratory virus season in the northern hemisphere. These factors suggest that use of $CO_2$ level ranges normalized to occupancy, with the above caveats, may be a practical method to estimate ventilation adequacy in schools than $CO_2$ derived ACH measures.

Many have suggested that $CO_2$ monitoring could be used as a surrogate for adequate ventilation [27, 54–58]. While the primary goal of this work was to provide an estimate of the correlation between SARS-CoV-2 infections and room ventilation in an IDD school, elevated $CO_2$ concentrations on their own may decrease cognitive task performance [37, 39–42, 59], concentration [60], increase cough and rhinitis [61], and teacher's reporting of health symptoms (e.g. fatigue, headache) [32]. Especially in schools for IDD students, improving ventilation, and achieving recommended $CO_2$ levels is likely to be important for school attendance [62], learning, and cognitive task performance [63, 64]. Monitoring $CO_2$ levels would provide real-time data in classrooms, and could be used to motivate interventions to improve ventilation and decrease $CO_2$ levels, from simple measures such as opening windows when possible to large-scale upgrades of school HVAC systems. Such measures should, however, be correlated with more rigorous measures of ventilation to account for numerous factors important for interpretation of the results [53]. Finally, it is important to note that we did not measure individual exposure over time to $CO_2$ concentrations, but rather concentrations in rooms. Individuals generally move between classrooms, therapy rooms, the outdoors, and other school areas, although less so during COVID.

This study also highlights the potential trade-offs in terms of improving in-line filtration and ventilation of particles and droplets to prevent infection and the potential expense required to upgrade HVAC systems in many schools [49]. Circulating air through a higher resistance particulate filter may not be possible for many older HVAC systems, requiring an expensive upgrade. In addition, filters may decrease ventilation rates, raising ambient $CO_2$, particulate, and volatile organic compound concentrations. Increasing the percentage of outdoor air brought in creates a trade-off between improved ventilation and the need to heat, cool, and condition the humidity of the incoming fresh air, increasing energy costs [37, 65]. Similarly, for IDD schools, measures such as opening windows create another trade off between improved ventilation and student safety. Given the vintage of most public school buildings and HVAC systems, as well as the budgets available for upgrades and remediation measures, this issue may benefit from more public funding and policy discussions.

One goal of community engaged research partnerships is to have direct impact on improving outcomes highly valued by the community. The comprehensive sampling in 100 rooms across 3 buildings revealed a wide variation in ventilation rates, as well as number of hours exposed to $CO_2$ levels greater than 1,000 ppm. Only 8 rooms had levels of $CO_2 \geq 2,000$ ppm for more than 30–130 minutes over the course of a school day, and of those 7 of 8 rooms were in the same building. These findings were subsequently validated by an independent HVAC contractor engaged by the school. Of note, the rooms with ventilation served by HVAC systems with MERV-13 filters had lower rates of PCR test positives for SARS-CoV-2. Overall,

these results have sparked discussion about upgrade of the ventilation systems. The school continues to seek state and federal funding in an on-going effort to ensure and exceed the ventilation standards recommended for schools.

## Conclusions

There was a statistically significant correlation between rooms with $CO_2$ levels ≥1,000 ppm and SARS-CoV-2 cases in an IDD school. Rooms served by HVAC systems with in-line MERV-13 filters had a lower incidence of SARS-CoV-2-positive PCR tests. This research partnership identified areas for improving in-school ventilation and the use of in-line, high efficiency air filtration.

## Supporting information

**S1 Fig. ACH curve fitting results for each room in the study.**
(PDF)

**S1 File. A CSV file containing data with room characteristics, including building, square footage and volume, occupancy, $CO_2$ concentration times, SARS-CoV-2 PCR positive subjects per room.**
(CSV)

**S2 File. A CSV file containing the time series $CO_2$ concentration data from each room used to derive the ACH estimates.**
(CSV)

## Acknowledgments

The authors would like to thank the Mary Cariola Center school community for their enthusiastic support and extensive partnership in this work. The authors would also like to thank the technologists of the UR Medicine Central Laboratory for their rapid performance of SARS-CoV-2 PCR tests. We are very grateful for Dr. Philip K. Hopke's (University of Rochester) expert guidance on the American Society of Heating, Refrigerating and Air-Conditioning Engineers (ASHRAE) standards for schools, $CO_2$ measurements and ACH calculations, and his critical reading of the manuscript. The inspiration for $CO_2$ measurements came from the outstanding social media videos of Dr. Katrine Wallace (University of Illinois; @epidemiologistkat). Research reported in this Rapid Acceleration of Diagnostics—Underserved Populations (RADx-UP) publication was supported by the National Institutes of Health under OT2 HD107553. This work was also supported by the University of Rochester Intellectual and Developmental Disabilities Research Center (P50 HD103536) from the NIH Eunice Kennedy Shriver Institute of Child Health and Human Development, and the University of Rochester Clinical and Translational Science Award (UL1 TR002001) from the National Center for Advancing Translational Sciences of the National Institutes of Health. The content is solely the responsibility of the authors and does not necessarily represent the official views of the National Institutes of Health.

## Author Contributions

**Conceptualization:** Martin S. Zand, Samantha Spallina, Alexis Ross, Karen Zandi.

**Data curation:** Martin S. Zand, Samantha Spallina, Alexis Ross.

**Formal analysis:** Martin S. Zand, Samantha Spallina, Alexis Ross, Anne Pawlowski, Christopher L. Seplaki, Jonathan Herington, Anthony M. Corbett, Kimberly Kaukeinen, Lisette Alcantara, Andrew Cameron.

**Funding acquisition:** Martin S. Zand.

**Investigation:** Martin S. Zand, Anne Pawlowski.

**Methodology:** Martin S. Zand, Samantha Spallina, Alexis Ross, Anne Pawlowski, Christopher L. Seplaki, Anthony M. Corbett, Kimberly Kaukeinen, Jeanne Holden-Wiltse, Dongmei Li, Andrew Cameron.

**Project administration:** Martin S. Zand, Alexis Ross, Karen Zandi, Nicole Beaumont.

**Resources:** Martin S. Zand, Anthony M. Corbett, Kimberly Kaukeinen, Jeanne Holden-Wiltse.

**Software:** Martin S. Zand, Kimberly Kaukeinen, Jeanne Holden-Wiltse.

**Supervision:** Martin S. Zand, Karen Zandi, Edward G. Freedman.

**Validation:** Martin S. Zand, Samantha Spallina, Alexis Ross, Anne Pawlowski, Anthony M. Corbett, Kimberly Kaukeinen.

**Visualization:** Martin S. Zand, Jonathan Herington, Anthony M. Corbett.

**Writing – original draft:** Martin S. Zand.

**Writing – review & editing:** Samantha Spallina, Alexis Ross, Karen Zandi, Anne Pawlowski, Christopher L. Seplaki, Jonathan Herington, Anthony M. Corbett, Kimberly Kaukeinen, Jeanne Holden-Wiltse, Edward G. Freedman, Dongmei Li, Andrew Cameron, Nicole Beaumont, Ann Dozier, Stephen Dewhurst, John J. Foxe.

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
