## [Decision Letter · Decision Letter 0]

31 Oct 2023

PONE-D-23-26827Ventilation during COVID-19 in a school for students with intellectual and developmental disabilities (IDD).PLOS ONE

Dear Dr. Zand,

Thank you for submitting your manuscript to PLOS ONE. After careful consideration, we feel that it has merit but does not fully meet PLOS ONE’s publication criteria as it currently stands. Therefore, we invite you to submit a revised version of the manuscript that addresses the points raised during the review process.

ACADEMIC EDITOR:  After taking into account the comments and feedback from all the reviewers, my overall decision is Minor Revisions. The paper presents the findings of a novel research work that was conducted. Overall, the paper is well-written. However, the reviewers have recommended a few minor changes related to the presentation of the Literature Review, Methodology, and Results. It is suggested that the authors review the comments and feedback from the reviewers and make the necessary revisions to the paper. 

We look forward to receiving your revised manuscript.

Kind regards,

Nirmalya Thakur

Academic Editor

PLOS ONE

Journal Requirements:

2. Please include a copy of Table 6-1 which you refer to in your text on page 7.

Reviewers' comments:

Reviewer's Responses to Questions

**Comments to the Author**

1. Is the manuscript technically sound, and do the data support the conclusions?

Reviewer #1: Yes

Reviewer #2: Yes

Reviewer #3: Yes

2. Has the statistical analysis been performed appropriately and rigorously? 

Reviewer #1: Yes

Reviewer #2: Yes

Reviewer #3: Yes

3. Have the authors made all data underlying the findings in their manuscript fully available?

Reviewer #1: Yes

Reviewer #2: Yes

Reviewer #3: Yes

4. Is the manuscript presented in an intelligible fashion and written in standard English?

Reviewer #1: Yes

Reviewer #2: Yes

Reviewer #3: Yes

5. Review Comments to the Author

Reviewer #1: The research article indicates the study examined the correlation of classroom ventilation (air exchanges per hour (ACH)) and exposure to CO2 ≥1,000 ppm with the incidence of SARS-CoV-2 over a 20-month period in a specialized school for students with intellectual and developmental disabilities (IDD).

Different ways of mitigating this issue is well explained.

Statistical analysis and the figures are depicted appropriately.

further the experimentation as per the research is accomplished for the set purpose.

the conclusion drawn is structured and provides some evidential inferences.

Henceforth the research article is strongly accepted for publishing

Reviewer #2: This article (Ventilation during COVID-19 in a school for students with intellectual and developmental disabilities (IDD)) investigates SARS-CoV-2 transmission in an IDD school and its relationship with ventilation and CO2 levels. Findings reveal that less than 20% of staff and students had PCR-confirmed SARS-CoV-2 infections, consistent with previous school studies. CO2 monitoring is highlighted as a potential tool to assess ventilation adequacy, with elevated CO2 levels impacting cognitive performance and health. The study emphasizes the need for improved ventilation and air filtration in schools and discusses the associated trade-offs, particularly for IDD-specialized schools. Public funding and policy discussions are suggested to address these challenges and enhance safety in educational settings. However, the authors could address the following suggestions for improving the quality of the article:

Abstract:

i. Well-structured and informative.

ii. Could provide more context on the importance of increased ventilation and the relevance of MERV13 filters.

Introduction:

Clarity and Context:

i. Effectively sets the stage for the study.

ii. It explains the importance of ventilation in reducing the risk of SARSCoV2 transmission in schools, especially for vulnerable populations with IDD.

Literature Review:

i. References prior studies and guidelines related to ventilation in school settings.

ii. Could include more recent research on SARSCoV2 transmission in schools.

Specific Aims and Research Gap:

i. Could benefit from explicitly stating the research aims.

ii. Needs a clearer statement of the research gap in the existing literature regarding ventilation and SARSCoV2 in IDD specialized schools.

NIH RADxUP Program:

i. Introduces the NIH RADxUP program.

ii. Might benefit from a brief description of the project's objectives within this context.

Study Scope:

i. Clearly outlines the scope, including the number of buildings and rooms analyzed.

ii. Specifies the target population (children and young adults with IDD) and age range.

Methods:

Human Subjects Protection:

i. Approved by the University of Rochester Medical Centre Research Subjects Review Board.

ii. Informed consent obtained from participants and parents or guardians for minors.

Room and Building Data:

i. Detailed description of the school's three different buildings and the selection of 100 school rooms.

ii. Metadata on room characteristics is included.

iii. Information about HVAC systems and their significance would be helpful.

SARSCoV2 Case Data:

Defined the timeframe for data collection but could include the rationale.

CO2 Measurements:

i. Explains the CO2 measurement process clearly.

ii. The use of Aranet sensors, their placement, and data recording intervals are explained.

iii. More information about the rationale for the specific measurement date might be beneficial.

Air Changes per Hour Estimation:

i. Describes the estimation of ACH well.

ii. A visual aid (Fig 1) is beneficial.

Statistical Analysis:

i. Details on the statistical analysis are provided.

ii. Appropriate use of Spearman's rank correlation test.

Results:

Room Characteristics:

i. Description of different buildings and HVAC systems.

ii. Explanation of the significance of HVAC systems and MERV filters might enhance understanding.

CO2 Level Profiles:

i. Patterns in CO2 concentration measurements are described.

ii. Could benefit from explaining the significance of these patterns.

Room CO2 Concentrations:

i. Rooms categorized based on CO2 levels during a school day.

ii. Explanation of the chosen CO2 ranges and their relevance is suggested.

Peak CO2 Levels by Room Functional Type and Building:

i. Presentation of peak CO2 levels by room type and building is provided.

ii. Context about rooms with wider CO2 level distributions might be valuable.

Temperature versus CO2 Relationship:

i. Temperature and CO2 relationships in different rooms are detailed.

ii. Some interpretation or discussion regarding temperature fluctuations would enhance this section.

Air Changes per Hour Estimates:

i. Distribution of ACH values and the comparison with recommended ACH levels are presented.

ii. Implications for ventilation could be discussed.

Incidence of SARSCoV2Positive PCR Tests:

i. Data on SARSCoV2 cases is presented.

ii. Discussion of the practical implications is needed.

Discussion:

Overall Interpretation:

i. Begins by discussing the limited number of SARSCoV2 cases and relates them to other studies.

ii. Should state the primary objective of the study more explicitly.

Interpretation of Findings:

i. Needs to clarify the practical significance of not achieving an ACH ≥ 4.

ii. The caveats related to using CO2 to estimate ACH need to be concise.

CO2 Monitoring:

i. Emphasizes the potential of CO2 monitoring.

ii. Needs more explicit recommendations for using CO2 monitoring in schools.

iii. Suggests discussing the thresholds for CO2 levels.

iv. Could elaborate on the connection between students' ability to tolerate masking and CO2 levels.

Ventilation and Air Filtration:

i. Discusses trade-offs between filtration and ventilation.

ii. Could be more concise and structured.

iii. The part about energy costs and outdoor air could be more concise.

Community Engagement and Policy Implications:

i. Discusses the impact on community engagement and potential policy implications.

ii. Might benefit from specific examples or proposals.

iii. Consider elaborating on how the research partnership influenced the school.

Conclusions:

i. Conclusions are clear.

ii. A summary of key findings and implications is suggested.

Reviewer #3: Authors investigated the correlation of classroom ventilation and exposure to CO2 with the incidence of SARS-CoV-2 over a 20-month period in a specialized school for students with intellectual and developmental disabilities. Study has deeply examined the strategies to keep the vulnerable populations of students in school while mitigating the risk of exposure and in-school spread of SARS-CoV-2 among students and staff. I appreciate the authors for studying such novel aspects in current scenario. Methodology, results and the discussion of the results are impressive. However, there are some minor corrections I suggest to improve the manuscript technically and scientifically sound. Highlight the research questions and key objectives (point-wise) at the end of the introduction part. Bring some more significant outcomes of the research that creates interest among the readers and include the future scope of the study in conclusion section. Authors may also recommend or suggest some significant points to the policy makers/decision makers. Add some statistical bars (error bars or standard deviation) in the figures if possible.

6. PLOS authors have the option to publish the peer review history of their article (what does this mean?). If published, this will include your full peer review and any attached files.

Reviewer #1: No

Reviewer #2: **Yes: **Dr.RAJKUMAR.S.C

Reviewer #3: No

---

## [Author Response · Author response to Decision Letter 0]

6 Dec 2023

Response to Reviewers: PONE-D-23-26827: Ventilation during COVID-19 in a school for students with intellectual and developmental disabilities (IDD) 

N.B. Please note that all line references [e.g. L25-31] refer to lines in the marked up PDF, not the clean copy.

Reviewer #1: The research article indicates the study examined the correlation of classroom ventilation (air exchanges per hour (ACH)) and exposure to CO2 ≥1,000 ppm with the incidence of SARS-CoV-2 over a 20-month period in a specialized school for students with intellectual and developmental disabilities (IDD). Different ways of mitigating this issue is well explained. Statistical analysis and the figures are depicted appropriately. Further the experimentation as per the research is accomplished for the set purpose. the conclusion drawn is structured and provides some evidential inferences. Henceforth the research article is strongly accepted for publishing.

We thank reviewer #1 for their comments, which do not recommend any changes.

N.B. Reviewer #2 included over 100 separate bullet points, a number of which were comments and not requests for revisions. For clarity, we have consolidated all the points, and only included those below that requested changes in the manuscript.

Reviewer #2: This article (Ventilation during COVID-19 in a school for students with intellectual and developmental disabilities (IDD)) investigates SARS-CoV-2 transmission in an IDD school and its relationship with ventilation and CO2 levels. Findings reveal that less than 20% of staff and students had PCR-confirmed SARS-CoV-2 infections, consistent with previous school studies. CO2 monitoring is highlighted as a potential tool to assess ventilation adequacy, with elevated CO2 levels impacting cognitive performance and health. The study emphasizes the need for improved ventilation and air filtration in schools and discusses the associated trade-offs, particularly for IDD-specialized schools. Public funding and policy discussions are suggested to address these challenges and enhance safety in educational settings. However, the authors could address the following suggestions for improving the quality of the article:

Literature Review: Could include more recent research on SARSCoV2 transmission in schools.

The literature on SARS-CoV-2 in schools is, as noted, extremely sparse. We have updated the references to include recent references [L12-14 and L30-34].

Specific Aims and Research Gap: Could benefit from explicitly stating the research aims. Needs a clearer statement of the research gap in the existing literature regarding ventilation and SARSCoV2 in IDD specialized schools. 

We have revised the introduction to explicitly state the research aims [L12-14]. We have also added a statement that, to our knowledge, there is no literature regarding ventilation and SARS-CoV-2 in IDD Schools [L32-34] 

NIH RADxUP Program: Might benefit from a brief description of the project's objectives within this context. We have revised the manuscript to frame the project’s objectives. [L76-81]

Room and Building Data: Information about HVAC systems and their significance would be helpful. Significance of HVAC systems, in general, was provided in the original manuscript in the introduction [L16-30]. Information linked to each HVAC system was provided in Supplemental File S1, linked to each room [L99]. We have revised the manuscript to provide some framing information about HVAC systems and their significance in ventilation [L207-208; L211-212]

SARSCoV2 Case Data: Defined the timeframe for data collection but could include the rationale. Respectfully, we did not have a choice regarding timeframe. SARSCoV2 case data collection began when NY State mandated reporting for schools and the , and we selected a 1 year period after this. Given the nature of the pandemic, we would not have been able to accurately “select” a timeframe prospectively for data collection.

CO2 Measurements: More information about the rationale for the specific measurement date might be beneficial. We have revised the manuscript to include an explanation for the selected measurement date range [L109-110].

Room Characteristics: Explanation of the significance of HVAC systems and MERV filters might enhance understanding. As noted above, we have added explanations of HVAC systems and MERV filters [L165-169].

CO2 Level Profiles: Could benefit from explaining the significance of these patterns. We are unclear which patterns the reviewer is referring to: the time-varying patterns of the measurements, the pattern of distribution within the buildings. The variation in patterns is already discussed, with notes regarding the significance (drowsiness, adherence to recommended standards). Thus, we have not modified this section.

Room CO2 Concentrations: Explanation of the chosen CO2 ranges and their relevance is suggested. Respectfully, the manuscript provides references to the CO2 ranges [L189; L191; L194]. The CO2 ranges are, as referenced, those recommended by national standards in the US by ASRHE and OSHA, as noted in the caption for Figure 3, and the references. Nonetheless, we have duplicated this information in the text [L187-189]

Peak CO2 Levels by Room Functional Type and Building: Context about rooms with wider CO2 level distributions might be valuable. We respectfully note that this is beyond the scope of the manuscript. CO2 levels are highly complex, and depend on room volume, room occupancy, activity level, ventilation (both active and passive), seasonality, and a large variety of factors we intentionally did not measure. The focus of the manuscript was to estimate ventilation with real-world data, not to intricately model all the factors that might have contributed to CO2 levels.

Temperature versus CO2 Relationship: Some interpretation or discussion regarding temperature fluctuations would enhance this section. We have added to the existing discussion regarding temperature fluctuations [L214-215; L218-220; L222-223].

Air Changes per Hour Estimates: Implications for ventilation could be discussed. Thank you for your comment. This is most appropriately addressed in the Discussion section, rather than in Results, which we have done.

Incidence of SARSCoV2Positive PCR Tests: Discussion of the practical implications is needed. Thank you for your comment. This is most appropriately addressed in the Discussion section, rather than in Results, which we have done.

Overall Interpretation: Should state the primary objective of the study more explicitly. We have restated the primary objective [L268-270], which was also stated in the introduction. 

Interpretation of Findings: Needs to clarify the practical significance of not achieving an ACH ≥ 4. The caveats related to using CO2 to estimate ACH need to be concise. We have addressed the recommendations of an ACH>4. Respectfully, while we done some editing for conciseness [L288-304], the issue of using CO2 to estimate ACH is complex, with a robust literature. Our discussion provides a number of critical caveats that would be important for a school measuring CO2 levels to consider for accurate use of this method, and to not overstate the utility of CO2 level measurements.

CO2 Monitoring: Needs more explicit recommendations for using CO2 monitoring in schools. Suggests discussing the thresholds for CO2 levels. Could elaborate on the connection between students' ability to tolerate masking and CO2 levels. Explicit recommendations for monitoring CO2 levels in schools are controversial, as noted by ASHRE. We have revised the manuscript to discuss this [L316-329]. Respectfully, we are not clear what “connection between the students ability to tolerate masking and CO2 levels” is referring to. Regardless of masking, expired CO2 will be the same, as masks are permeable to CO2; Masking is a neutral intervention in terms of CO2 levels. Thus, we have not addressed this in the discussion. 

Ventilation and Air Filtration: Could be more concise and structured. The part about energy costs and outdoor air could be more concise. Respectfully, all the issues briefly raised are critical considerations in terms of tradeoffs. In particular, there is only single sentence about outdoor air heating and energy costs, which is already concise [L336-338]. We have left this as is.

Community Engagement and Policy Implications: Might benefit from specific examples or proposals. Consider elaborating on how the research partnership influenced the school. Respectfully, specific examples at the end of the study would greatly expand the scope of the paper, without hard data. Rather than adding anecdotal examples, this will be the subject of an additional manuscript.

Conclusions: A summary of key findings and implications is suggested. We have added this.

Reviewer #3: Authors investigated the correlation of classroom ventilation and exposure to CO2 with the incidence of SARS-CoV-2 over a 20-month period in a specialized school for students with intellectual and developmental disabilities. Study has deeply examined the strategies to keep the vulnerable populations of students in school while mitigating the risk of exposure and in-school spread of SARS-CoV-2 among students and staff. I appreciate the authors for studying such novel aspects in current scenario. Methodology, results and the discussion of the results are impressive. However, there are some minor corrections

I suggest to improve the manuscript technically and scientifically sound. Highlight the research questions and key objectives (point-wise) at the end of the introduction part. We have revised the manuscript accordingly [L12-14].

Bring some more significant outcomes of the research that creates interest among the readers and include the future scope of the study in conclusion section. We have revised the discussion section to emphasize significant outcomes and to discuss the future scope of potential research.

Authors may also recommend or suggest some significant points to the policy makers/decision makers. We have revised the discussion section to cautiously suggest some points to policy makers. We are very mindful that as a single school study, in a very distinct population, it likely does not rise to the level that would warranted basing 

Add some statistical bars (error bars or standard deviation) in the figures if possible. Respectfully, error bars were added when appropriate or relevant. Examples of CO2 concentration measurements over time are not averages, and thus error bars are not appropriate [Fig 1, 3, 6]. Similarly, figures that contain histograms counts would not have error bars [Fig 2A, 7A]. Points denoting mean minimum and maximum ACH have standard deviations of <5% for all estimates, and error bars on each of 100 points would make the figures unreadable. Supplemental Figure 2 does show mean and standard deviation for each ACH estimate.

---

## [Decision Letter · Decision Letter 1]

7 Feb 2024

PONE-D-23-26827R1Ventilation during COVID-19 in a school for students with intellectual and developmental disabilities (IDD).PLOS ONE

Dear Dr. Zand,

Thank you for submitting your manuscript to PLOS ONE. After careful consideration, we feel that it has merit but does not fully meet PLOS ONE’s publication criteria as it currently stands. Therefore, we invite you to submit a revised version of the manuscript that addresses the points raised during the review process.

We look forward to receiving your revised manuscript.

Kind regards,

Yanping Yuan

Academic Editor

PLOS ONE

Journal Requirements:

Reviewers' comments:

Reviewer's Responses to Questions

**Comments to the Author**

1. If the authors have adequately addressed your comments raised in a previous round of review and you feel that this manuscript is now acceptable for publication, you may indicate that here to bypass the “Comments to the Author” section, enter your conflict of interest statement in the “Confidential to Editor” section, and submit your "Accept" recommendation.

Reviewer #1: All comments have been addressed

Reviewer #2: All comments have been addressed

Reviewer #3: All comments have been addressed

2. Is the manuscript technically sound, and do the data support the conclusions?

Reviewer #1: Yes

Reviewer #2: Partly

Reviewer #3: Yes

3. Has the statistical analysis been performed appropriately and rigorously? 

Reviewer #1: Yes

Reviewer #2: Yes

Reviewer #3: Yes

4. Have the authors made all data underlying the findings in their manuscript fully available?

Reviewer #1: Yes

Reviewer #2: Yes

Reviewer #3: Yes

5. Is the manuscript presented in an intelligible fashion and written in standard English?

Reviewer #1: Yes

Reviewer #2: Yes

Reviewer #3: Yes

6. Review Comments to the Author

Reviewer #1: All the reviewers queries are addressed appropriately and hence recommend for publication.

A summary of key findings and implications is suggested and added.

The discussion section revised to cautiously suggest some points to policy makers

Error bars were added when appropriate or relevant. Similarly, figures that contain

histograms counts would not have error bars [Fig 2A, 7A]. Points denoting mean minimum and maximum

ACH have standard deviations of <5% for all estimates, and error bars on each of 100 points would make

the figures unreadable.

Reviewer #2: The article lacks chapter numbers throughout.

The figures lack titles and numbering, and the data within them contains three-line indications without corresponding labels for each line.

Additionally, the graphs utilize multiple colors for different lines. Please differentiate the graph lines using distinct styles.

Refer to the following transaction journal and include it in the reference section that is more suitable for this article

1. Covid-19: automatic detection of the novel coronavirus disease from ct images using an optimized convolutional neural network.

2. The role of internet of things to control the outbreak of COVID-19 pandemic

Please review and address these comments for further improvement.

Reviewer #3: Authors have tried their best to address all my comments. The article may be accepted for publication.

7. PLOS authors have the option to publish the peer review history of their article (what does this mean?). If published, this will include your full peer review and any attached files.

Reviewer #1: **Yes: **ROOPA B S

Reviewer #2: No

Reviewer #3: No

---

## [Author Response · Author response to Decision Letter 1]

13 Feb 2024

The response to reviewers is also submitted in a formatted document.

Response to Reviewers: PONE-D-23-26827: Ventilation during COVID-19 in a school for students

with intellectual and developmental disabilities (IDD)

Response to Reviewers 1 and 3: We thank the reviewers for having recommended accepting the manuscript with no further revisions. 

Response to Reviewer #2:

1. The article lacks chapter numbers throughout. Respectfully, the article was prepared according the PLoS One Instructions for Authors style guide, which does not require chapter numbers. Indeed, the sample manuscript body in the Instructions to Authors (, and the available templates provided on the web site, do not include this formatting. We have thus not changed the manuscript so that we remain within the format required by PLoS.

2. The figures lack titles and numbering, and the data within them contains three-line indications without corresponding labels for each line. 

The Figures are submitted per the instructions for authors: “Do not include author names, article title, or figure number/title/caption within figure files. That information will go into your figure caption in the manuscript”. Numbering is present in both the figure legends and in the upper right corner of the pages containing the figure itself in the reviewer’s copy. Thus, to remain within the format required by PLoS, we have not changed the figures to include numbers and titles.

3. Additionally, the graphs utilize multiple colors for different lines. Please differentiate the graph lines using distinct styles. 

The reviewer did not specify which graph is referred to, and so we have responded for each graph:

Figure 1 – Each panel has more than one line uses a different style: A) there are two lines in the graph, one solid with coloring explained in the figure legend, and the other dashed B) there is one solid line (fitted curve), and multiple data points. We have not changed the figure.

Figure 2 – Neither panel A or B contain “different lines” – A) is a histogram and B) is a scatter plot without a line. As the comment does not appear to apply to this figure, we have not changed it.

Figure 3 – Each panel with more than one line uses a different style: A) there are two lines in the graph, one solid with coloring explained in the figure legend, and the other dashed B) There are no lines, each is a scatter plot. We have not changed the figure.

Figure 4 – This is a stack graph, with no data lines. As the comment does not appear to apply to this figure, we have not changed it.

Figure 5 – There are three “lines” denoting the bands corresponding to CO2 level boundaries, clearly explained in the legend. All are dashed, and a corresponding color-coded bar on the y-axis serves as a second guide for the reader to the CO2 level bands. Using a distinct style for each boundary line would detract from the figure clarity. We have not changed the figure.

Figure 6 – These are time vs. value dot plots of the masurements. Given the measurement density, to plot them as lines that would be distinguishable by pattern (e.g. dashed, dot-dash, solid) would require significant down-sampling the data for smoothing, detracting from the accuracy and rigorous presentation of the data. We have not changed the figure.

Figure 7 – A is a histogram, and C is a biplot with a single superimposed line. We have changed panel C so that line style differs among the ACH levels, as reqeusted.

Figure 8 – Panel A is a box-plot and B is are biplots, Neither panel contains “different lines”.

As the comment does not appear to apply to this figure, we have not changed it.

4. Refer to the following transaction journal and include it in the reference section that is more suitable for this article

(a) Covid-19: automatic detection of the novel coronavirus disease from CT images using an optimized convolutional neural network.

(b) The role of internet of things to control the outbreak of COVID-19 pandemic

Respectfully, we are very confused by these comments. Our article does not refer at any point to medical imaging, the use of neural networks, or the internet of things. Is it possible that these comments might refer to a different manuscript than ours? If not, we would very much appreciate a clearer elucidation of these concerns.

---

## [Decision Letter · Decision Letter 2]

21 Feb 2024

Ventilation during COVID-19 in a school for students with intellectual and developmental disabilities (IDD).

PONE-D-23-26827R2

Dear Dr. Zand,

We’re pleased to inform you that your manuscript has been judged scientifically suitable for publication and will be formally accepted for publication once it meets all outstanding technical requirements.

Kind regards,

Yanping Yuan

Academic Editor

PLOS ONE

Additional Editor Comments (optional):

Reviewers' comments:

Reviewer's Responses to Questions

**Comments to the Author**

1. If the authors have adequately addressed your comments raised in a previous round of review and you feel that this manuscript is now acceptable for publication, you may indicate that here to bypass the “Comments to the Author” section, enter your conflict of interest statement in the “Confidential to Editor” section, and submit your "Accept" recommendation.

Reviewer #2: All comments have been addressed

2. Is the manuscript technically sound, and do the data support the conclusions?

Reviewer #2: Partly

3. Has the statistical analysis been performed appropriately and rigorously? 

Reviewer #2: Yes

4. Have the authors made all data underlying the findings in their manuscript fully available?

Reviewer #2: Yes

5. Is the manuscript presented in an intelligible fashion and written in standard English?

Reviewer #2: Yes

6. Review Comments to the Author

Reviewer #2: The authors have successfully addressed all the raised concerns and implemented the necessary revisions to the manuscript.

7. PLOS authors have the option to publish the peer review history of their article (what does this mean?). If published, this will include your full peer review and any attached files.

Reviewer #2: No

---

## [Editor Report · Acceptance letter]

22 Mar 2024

PONE-D-23-26827R2 

PLOS ONE

Dear Dr. Zand, 

I'm pleased to inform you that your manuscript has been deemed suitable for publication in PLOS ONE. Congratulations! Your manuscript is now being handed over to our production team.

Kind regards, 

on behalf of

Prof. Yanping Yuan 

Academic Editor

PLOS ONE